# Cannabinoids and Sleep: Exploring Biological Mechanisms and Therapeutic Potentials

**DOI:** 10.3390/ijms25073603

**Published:** 2024-03-22

**Authors:** Martina D’Angelo, Luca Steardo

**Affiliations:** Psychiatry Unit, Department of Health Sciences, University of Catanzaro Magna Graecia, 88100 Catanzaro, Italy; martina.dangelo001@studenti.unicz.it

**Keywords:** endogenous cannabinoid system, sleep, cannabidiol, palmitoylethanolamide

## Abstract

The endogenous cannabinoid system (ECS) plays a critical role in the regulation of various physiological functions, including sleep, mood, and neuroinflammation. Phytocannabinoids such as Δ9-tetrahydrocannabinol (THC), cannabidiol (CBD), cannabinomimimetics, and some N-acylethanolamides, particularly palmitoyethanolamide, have emerged as potential therapeutic agents for the management of sleep disorders. THC, the psychoactive component of cannabis, may initially promote sleep, but, in the long term, alters sleep architecture, while CBD shows promise in improving sleep quality without psychoactive effects. Clinical studies suggest that CBD modulates endocannabinoid signaling through several receptor sites, offering a multifaceted approach to sleep regulation. Similarly, palmitoylethanolamide (PEA), in addition to interacting with the endocannabinoid system, acts as an agonist on peroxisome proliferator-activated receptors (PPARs). The favorable safety profile of CBD and PEA and the potential for long-term use make them an attractive alternative to conventional pharmacotherapy. The integration of the latter two compounds into comprehensive treatment strategies, together with cognitive–behavioral therapy for insomnia (CBT-I), represents a holistic approach to address the multifactorial nature of sleep disorders. Further research is needed to establish the optimal dosage, safety, and efficacy in different patient populations, but the therapeutic potential of CBD and PEA offers hope for improved sleep quality and general well-being.

## 1. Endocannabinoid System

The endocannabinoid system (ECS) plays a pivotal role in regulating numerous biological functions, spanning neurodevelopment, cognition, mood, sleep, appetite, and reward [1,2]. This article examines the complex interaction between the ECS and sleep architecture, delving into the mechanisms behind cannabinoid modulation of sleep patterns and any spin-offs in clinical practice. Importantly, it consolidates findings from preclinical and clinical studies, illuminating the multifaceted pharmacological effects of cannabinoids and cannabinomimetics and their role in targeting various pathophysiological pathways implicated in sleep disturbances. Furthermore, it emphasizes the necessity for additional research to establish optimal dosing regimens, long-term safety, and efficacy across diverse patient populations. By incorporating cannabinoids into comprehensive treatment strategies alongside cognitive–behavioral therapy for insomnia (CBT-I) and lifestyle adjustments, this article advocates for a holistic approach to addressing the multifaceted nature of sleep disorders, thus presenting a promising pathway for improving the quality of life for millions worldwide.

Humans have historically utilized Cannabis sativa for its renowned effects, including euphoria, stress alleviation, enhanced appetite, and potential anxiety modulation. The isolation of its active component, Δ9-tetrahydrocannabinol (THC), took place in 1964 [3]. However, the major constituents of the ECS were not identified until the early 1990s, when it was widely recognized that the ECS consists of cannabinoid receptors (CBRs), endogenous ligands (such as 2-arachidonoylglycerol (2-AG), N-arachidonylethanolamine (AEA or anandamide), and enzymes responsible for synthesizing and degrading ECS [4,5,6,7,8]. The widespread distribution of the ECS in both central and peripheral locations suggests its involvement in various human physiological functions [9]. The ECSs are released ‘on demand’ from membrane phospholipid precursors. Although AEA synthesis can result from various metabolic pathways, a specific N-acylphosphatidylethanolamine phospholipase D (NAPE-PLD) is currently considered the main enzyme responsible for AEA production [10]. At the same time, a distinct phospholipase C followed by the activity of sn-1-diacylglycerol lipase (DAGL) leads to the synthesis of 2-AG [11]. Cellular uptake from extracellular to intracellular space is attributed to a putative ‘endocannabinoid membrane transporter (EMT)’ that probably uptakes both AEA and 2-AG. There is several experimental evidence to support the notion that AEA transport across membranes is mediated by proteins [12]. In addition to the G-protein, coupled cannabinoid receptors such as subtypes type 1 (CB1-R) and type 2 (CB2-R), the other “non-canonical” extended signaling networks of the ECS, include the receptors GPR55 and peroxisome proliferator-activated receptor-alpha (PPARα), inotropic cannabinoid receptors (TRP channels), protein transporters (FABP family), and other fatty acid derivatives [10,13]. Both CB1R and CB2R are G-protein-coupled receptors, exhibiting a diverse range of functions that include inhibiting adenylyl cyclase and calcium influx, activating potassium channels and various mitogen-activated protein kinases, and recruiting β-arrestin [14].

The activation of CB1 and CB2 receptors modulates intracellular pathways, influencing enzyme activity or gene expression [15,16,17,18]. CB1 receptors, particularly neuronal ones, regulate neurotransmitter release, impacting synaptic transmission [19,20,21] since, synthesized on demand in response to the increased intracellular calcium, they contribute to short- and long-term plasticity [22]. Historically, CB1R was primarily associated with expression in various brain regions, such as the cortex, cerebellum, hippocampus, and basal ganglia [23], while CB2R was predominantly found in immune cells. However, recent studies have challenged this classification, identifying new endocannabinoid binding sites in peripheral organs via CB1R [14,24]. Due to its widespread distribution in the central nervous system, particularly in presynaptic terminals, where it modulates transmitter release upon activation, CB1R plays a crucial role in regulating various neurological functions, including memory, cognition, emotion, motor control, feeding, sleeping, and pain perception [14]

Moreover, it is now recognized that AEA and 2-AG are not the exclusive endogenous substances responsible for activating the ECS. Various bioactive compounds sharing structural resemblances have been discovered, each capable of engaging cannabinoid receptors. These include N-acylethanolamines (NAEs), like N-palmitoylethanolamide (PEA), N-oleoylethanolamide (OEA), N-stearoylethanolamide, and N-linoleoylethanolamide, alongside lipid mediators from the acylglycerol family, such as 2-palmitoylglycerol (2-PG) and 2-oleoylglycerol (2-OG) [25]. 

## 2. Endocannabinoids and Sleep

Emerging research indicates that 2-AG plays a pivotal role in regulating sleep patterns through the circadian rhythm. Notably, studies have revealed heightened levels of 2-AG in sleep-deprived healthy young adults [26]. Furthermore, Soni et al. (2017) demonstrated the presence of endogenous cannabinoids, particularly 2-AG and AEA, within the cholinergic laterodorsal tegmentum (LDT), influencing cortical gamma band activity associated with states of arousal, suggesting a role for 2-AG in arousal modulation [27].

Numerous in vivo investigations have delved into the role of 2-AG in sleep regulation. For instance, Perez-Morales and colleagues (2013, 2014) found that direct administration of 2-AG into the lateral hypothalamus of mice increased the duration of REM sleep, while the administration of a CB1 receptor antagonist elicited opposing effects, indicating that CB1 receptor activation by 2-AG induces sleepiness [28,29]. Moreover, Perez-Morales et al. (2012) discovered that tetrahydrolipstatin, an inhibitor of the 2-AG-synthesizing enzyme DAGL, reduces the activity of melanin-concentrating hormone (MCH)-releasing neurons in the hypothalamus, resulting in decreased REM sleep and food intake [30,31]. This suggests that CB1 receptors may regulate sleep by modulating the activity of MCH neurons, predominantly located in the lateral hypothalamus, which is active during the REM stage of sleep. Similar to 2-AG, AEA levels are also influenced by the circadian rhythm. Vaughn et al. (2010) found that AEA levels were three times higher immediately after waking compared to before sleep [32]. Extensive in vivo studies have demonstrated that AEA administration in rats increases slow-wave sleep (SWS) and non-rapid eye movement (NREM) sleep while inhibiting wakefulness [33,34]. Microinjections of AEA in rats have been found to increase adenosine levels in the basal forebrain, consequently increasing SWS [35]. Additionally, Dasilva et al. (2014) showed that AEA activation of CB1 receptors in the thalamus induces Δ-like oscillations characteristic of deep NREM sleep, possibly through decreased cortical feedback by the dorsal lateral geniculate nucleus. FAAH-knockout mice exhibited increased SWS along with elevated extracellular FAAH concentration [36].

In 2016, Murillo-Rodríguez et al. investigated the effects of two synthetic cannabinoids, URB597 (FAAH inhibitor) and VDM-11 (anandamide membrane transporter inhibitor), in sleep-deprived rats. Administration of VDM-11 led to sleep rebound, while URB597 exhibited opposing effects. Furthermore, administration of VDM-11 promoted SWS and NREM sleep via activation of the paraventricular thalamic nucleus, pedunculopontine tegmental nucleus, and the anterior hypothalamic area [37]. These studies collectively suggest that AEA is a crucial modulator of sleep via CB1 receptor activation, wherein heightened AEA concentration is associated with increased sleep duration.

## 3. Phytocannabinoids and Synthetic Cannabinomimetics

The term cannabinoids, or phytocannabinoids, commonly refers to approximately 120 or more compounds identified in the trichomes of Cannabis sativa. Among them, Δ9-tetrahydrocannabinol (Δ9-THC or THC), cannabidiol (CBD), cannabichromene (CBC), and cannabigerol (CBG) are the most prevalent. Cannabinoids exert their effects within the body by modulating multiple targets simultaneously. [38]. For instance, Δ9-THC, the mind-altering element of Cannabis, functions as a partial agonist of the cannabinoid receptors CB1 and CB2, whereas CBD, the second most prevalent phytocannabinoid, shows a limited affinity for CB1 or CB2 receptors, despite having the same molecular formula as Δ9-THC.

As a result, numerous additional non-cannabinoid receptors are involved, occasionally serving as the principal players in the mechanism of action of phytocannabinoids. [39]. These include members of large G-protein-coupled receptors (GPCRs) such as GPR55, GPR18, GPR3, GPR6, and GPR12. Moreover, ligand-operated ion channels, transient receptor potential channels (TRP), and intracellular receptors such as peroxisome proliferator-activated receptors (PPARs) have been demonstrated to be involved in the biological effects of cannabinoids [40].

The complex pharmacological characteristics of phytocannabinoids remain only partially comprehended, and are still a subject of interest in the quest for innovative treatments against diverse human ailments. Apart from their well-known benefits in alleviating chemotherapy-related symptoms, cannabinoids are increasingly used to treat medical conditions including chronic pain, cancer, anxiety, insomnia, and epilepsy [41].

Ultimately, the diverse realm of cannabinoids encompasses numerous synthetic compounds engineered to emulate the activity of Δ9-THC and/or endocannabinoids by directly modulating cannabinoid receptors CB1 and CB2 [42]. This category of molecules encompasses synthetic potent agonists of CB1 such as AM-1235, arachidonic-2′ chloroethylamide (ACEA), JWH-073, and methanandamide, as well as synthetic potent agonists of CB2 like JWH133, JWH015, HU-308, and AM1241. Mixed CB1/CB2 agonists such as WIN55,212-2, HU-210, and CP55,940, and selective antagonists such as Rimonabant/SR141716 and AM251 for CB1, as well as SR144528 and AM630 for CB2, are also part of this category [43]. Moreover, cannabimimetic agents encompass activators or inhibitors of fatty acid amide hydrolase (FAAH) and monoacylglycerol lipase (MAGL), the two primary enzymes responsible for the breakdown of AEA and 2-AG, as well as agents that indirectly modify endocannabinoid synthesis and turnover [44].

## 4. N-Acylethanolamides

N-acylethanolamides (NAEs) constitute another class of bioactive lipids closely associated with endocannabinoids (eCBs), although they do not engage with cannabinoid receptors; thus, they are not classified as eCBs. Noteworthy among them are OEA and PEA. Both OEA and PEA are synthesized by similar enzymatic pathways responsible for AEA synthesis. OEA primarily undergoes hydrolysis by FAAH, while PEA is predominantly metabolized by N-acylethanolamine-hydrolyzing acid amidase (NAAA) [45,46]. The subsequent sections will delve into these molecules and their implications for sleep patterns.

Diurnal fluctuations in the synthesis of OEA and PEA appear to be tightly regulated across various brain regions, including the pons, hypothalamus, and hippocampus. Notably, both OEA and PEA exhibit peak levels during the dark phase of the cycle, as highlighted in the study by Murillo-Rodriguez et al. (2006), suggesting a potential involvement in promoting wakefulness [47]. Subsequent investigations have delved into understanding the impact of OEA on sleep following systemic or intra-lateral hypothalamic administration, as elucidated by Soria-Gómez et al. (2010). The findings revealed that OEA administration led to a reduction in overall sleep duration, particularly affecting REM sleep, regardless of the route of administration. Additionally, infusion of OEA into the lateral hypothalamus resulted in diminished c-Fos expression within this region [48], although the precise neurotransmitter modality of these affected cells remains unexplored.

Similarly, research on PEA has highlighted its capacity to enhance wakefulness upon administration into the lateral hypothalamus or the dorsal raphe nucleus of rats, as reported by Murillo-Rodriguez et al. (2011) [49]. This enhancement came at the expense of both NREM and REM sleep stages, further underscoring the arousing effects of PEA within the central nervous system. Additional evidence supporting the role of OEA in promoting wakefulness comes from a study involving healthy volunteers (age range: 19–49, 12 women and 8 men), where levels of OEA increased in the cerebrospinal fluid (CSF) following 24 h of sleep deprivation. Interestingly, the measurements of AEA remained unchanged during this period [50]. Furthermore, research indicates that plasma levels of OEA are elevated in patients with obstructive sleep apnea (OSA), and these levels positively correlate with both the respiratory distress index and the body mass index [51].

The impact of PEA on the modulation of sleep–wake rhythm is of considerable interest. PEA, an endogenous fatty acid amide, has attracted significant attention due to its potential role in regulating sleep patterns through complex interactions with the ECS and peroxisome proliferator-activated receptor alpha (PPAR-α). Notably, PEA shares a mechanism with oleamide concerning PPAR-α.

Within the ECS, PEA exerts multifaceted actions, including enhancing anandamide-mediated signaling, a crucial regulator of sleep–wake cycles. Preclinical studies employing animal models have provided compelling evidence supporting PEA’s efficacy in regulating sleep. For instance, Guida et al. (2015) demonstrated that PEA supplementation induced microglial changes associated with increased migration and phagocytic activity, suggesting its potential to modulate neuroinflammatory processes linked to sleep disturbances [52]. Additionally, Keppel Hesselink and Hekker (2012) reported improvements in sleep quality and duration following PEA administration in neuropathic pain models, indicating broader therapeutic potential for sleep disorders [53]. The interaction between PEA and PPAR-α further underscores its significance in sleep regulation. PPAR-α emerges as a key regulator of sleep, exerting its effects through multifaceted interactions with various molecular pathways. Within the central nervous system, PPAR-α is abundantly expressed in key brain regions implicated in sleep–wake regulation, such as the hypothalamus and brainstem nuclei [54]. Activation of PPAR-α has been associated with alterations in sleep architecture, particularly characterized by enhancements in SWS and improvements in sleep consolidation [55], essential for physiological and cognitive functioning.

Moreover, PPAR-α’s involvement extends to the regulation of circadian rhythms, influencing the timing and duration of sleep phases. Research suggests that PPAR-α may interact with core components of the circadian clock machinery, modulating the expression of clock genes and thus impacting the sleep–wake cycle [56]. Additionally, PPAR-α activation has been linked to the regulation of energy metabolism, particularly lipid metabolism, indirectly influencing sleep patterns through metabolic signaling pathways [55]. Evangelista et al. (2018) elucidated PEA’s anti-inflammatory properties mediated through PPAR-α activation, potentially contributing to its efficacy in managing sleep disorders associated with neuroinflammation [57]. Furthermore, Rao et al. (2021) highlighted the role of PEA in modulating sleep through its interactions with PPAR-α, offering insights into the molecular mechanisms underlying its effects on sleep [58].

Clinical studies have further validated the therapeutic potential of PEA in sleep medicine. Evangelista et al. (2018) conducted clinical trials evaluating the efficacy of PEA supplementation in improving sleep parameters in patients with various pathological conditions [57]. The results showed promising outcomes, with significant enhancements in sleep quality and duration observed across different patient cohorts. These findings underscore the translational relevance of preclinical studies, emphasizing the potential clinical utility of PEA in managing sleep disorders. PEA presents itself as a promising therapeutic agent for sleep regulation, acting through interactions with the ECS and PPAR-α. Preclinical studies using animal models have provided valuable insights into the mechanisms underlying PEA’s effects on sleep, while clinical trials have demonstrated its efficacy in improving sleep parameters in human subjects. Continued research into the molecular mechanisms and clinical applications of PEA holds significant promise for the development of novel therapeutic interventions for sleep disorders.

## 5. Research on Cannabinoid and Sleep

While various studies have suggested that chronic exposure to THC and other CB1-activating compounds may lead to slight improvements in sleep, recent meta-analyses, such as one conducted by Mucke et al. (2018) based on Cochrane data, have critiqued the quality of these findings [59]. Furthermore, a critical review of clinical trial literature by Kuhathasan et al. (2019) emphasized the necessity of additional large-scale controlled clinical trials to ascertain the efficacy of such interventions [60]. Nevertheless, despite these methodological concerns, a significant number of cannabis users report improvements in sleep quality as a primary motivator for use [61]. Moreover, an increasing body of evidence suggests that cannabis usage may offer therapeutic benefits to individuals experiencing poor sleep quality associated with conditions such as post-traumatic stress disorder (PTSD) [62,63,64,65,66] and pain [67,68,69,70,71].

A study involving 27 healthy participants found that the acute administration of a 300 mg dose of CBD did not produce any significant changes in polysomnography or subjective sleep measures when compared to a placebo [72]. In another crossover study, eight individuals were randomly assigned to four different treatment conditions: Δ9-THC alone (at a dose of 15 mg); a combination of Δ9-THC (5 mg) with CBD (5 mg); a combination of Δ9-THC (15 mg) with CBD (15 mg); and a placebo, administered via an oromucosal spray [73].

The results indicated no notable alterations in polysomnography with ΔTHC alone; however, there were slight reductions in stage N3 sleep observed with both the Δ9-THC and CBD combination doses. Research into cannabinoid interventions for insomnia disorder remains limited. An early randomized controlled trial (RCT) investigated the effects of single-dose CBD at varying concentrations (40, 80, and 160 mg) compared to both placebo and nitrazepam (5 mg) in a small sample of individuals with insomnia. Notably, this study employed single items from an unvalidated sleep scoring test. Interestingly, all doses of CBD were found to decrease dream recall, while the 160 mg CBD dose demonstrated a notable increase in subjective sleep duration [74].

Conducted with a precise design, a more recent investigation has scrutinized the safety and effectiveness of ZTL-101, a carefully formulated blend of cannabinoids comprising Δ9-THC 10 mg, CBN 1 mg, and CBD 0.5 mg, in contrast to a placebo among a cohort of 23 individuals suffering from chronic insomnia disorder. Over a duration of 2 weeks, participants self-administered ZTL-101 one hour before bedtime, with the option to double the dosage after the fourth night (a decision made by 52% of participants). The results revealed a substantial improvement in subjective sleep quality, with a noteworthy increase in points on the Insomnia Severity Index. Additionally, ZTL-101 demonstrated enhancements in self-reported sleep-onset latency, total sleep time, subjective sleep quality, and the sensation of rejuvenation upon awakening [75].

Moreover, emerging research suggests that nabilone, a synthetic THC analog, holds promise for addressing nightmares in individuals with PTSD. Notably, both a randomized controlled trial (RCT) and an open-label study have demonstrated a significant reduction in nightmare frequency with nabilone administration, ranging from 0.5 mg/d to a maximum of 3 mg/d. Mild adverse effects, including dry mouth, headache, and dizziness, were reported in approximately half of the patients. However, it is important to acknowledge the limitations of the study by Fraser et al. [76], such as the small sample size in the RCT and the open-label design.

To bolster these findings, further well-designed trials involving larger and more diverse clinical populations, including females and individuals with non-trauma-related nightmare disorders, are warranted. Additionally, longer-term follow-up is essential to assess the sustained efficacy and safety of nabilone use. Despite the promising initial results regarding nabilone’s efficacy in reducing PTSD-related nightmares, uncertainties remain regarding the long-term safety of CB1 receptor agonists in this population, particularly considering their complex comorbidities and the heightened risk of substance abuse [77].

CBD and THC constitute the primary components of cannabis extracts, which are increasingly employed in clinical applications for mitigating several neuropsychiatric diseases. CBD has been noted for its reported antidepressant and anxiolytic properties [78,79], with emerging evidence suggesting its efficacy in addressing neuropsychiatric conditions such as Parkinsonism [80], epilepsy [81], Tourette’s syndrome [82], schizophrenia [83], post-traumatic stress disorder [84], and autism spectrum disorder [85]. Recent research also hints at the potential therapeutic benefits of CBD in sleep disorders, including obstructive sleep apnea, narcolepsy, REM sleep behavior disorder, and PTSD-related nightmares [86,87,88]. Approved in many countries under the brand name Epidiolex, CBD is utilized in treating Dravet syndrome and Lennox–Gastaut syndrome, severe forms of childhood epilepsy that are resistant to conventional treatments [89,90].

Despite CBD’s low affinity for both CB1 and CB2 receptors [91], it interacts with CB1 receptors as a negative allosteric modulator [92]. Recent studies [93] have indicated that CBD functions as an inverse agonist of the CB2 receptor, potentially elucidating its anti-inflammatory properties given the expression of CB2 receptors in immune cells, with inhibition likely leading to reduced inflammation [94]. Unlike THC, CBD lacks psychoactive properties, rendering it valuable for the treatment of various disorders. The impact of CBD on sleep regulation has attracted interest, with low doses promoting sleep and high doses inducing sedation [73]. This effect may arise from CBD’s capacity to lower cortisol levels by downregulating the corticotropin-releasing hormone (CRH) gene, thereby fostering sedation, as corticosteroids enhance wakefulness [95].

In 2021, Murillo-Rodriguez et al. discovered that chronic CBD injections (5 or 30 mg/kg) in adolescent mice heightened alertness, decreased REM sleep during the lights-on period, and enhanced SWS while diminishing wakefulness during the lights-off period. Elevated NeuN expression in the suprachiasmatic nucleus (SCN) suggests the involvement of the SCN in CBD-induced sleep modulation [96]. Moreover, CBD (75–300 mg) significantly enhanced sleep satisfaction among patients with Parkinson’s disease in a phase II/III double-blind, placebo-controlled clinical trial [97]. 

THC serves as the primary psychoactive constituent within the cannabis plant and holds significant therapeutic promise across various medical domains [98]. Additionally, a synthetic analog of THC, nabilone, was employed successfully to mitigate chemotherapy-induced emesis and nausea [99], and was recently found to ameliorate sleep disturbances in individuals with Parkinson’s Disease [100]. Despite its limited impact on nocturnal sleep, THC administration has been associated with adverse effects such as increased somnolence, reduced sleep onset latency, mood fluctuations, and cognitive impairment persisting into the subsequent day [73]. Furthermore, scientific discourse highlights THC’s intricate role in modulating the sleep–wake cycle, with empirical studies elucidating its capacity to facilitate SWS and modulate the duration of REM sleep alongside associated ocular activity [101,102].

## 6. Comments and Future Perspectives

Presently, a burgeoning body of research underscores the integral role of the endocannabinoid system in regulating the sleep–wake cycle. However, despite this understanding, the clinical application of cannabinoid or cannabinomimetic therapies for sleep disorders lacks substantial support due to limited published studies and prevalent biases in both clinical and preclinical investigations. To advance our comprehension, concerted scientific endeavors are imperative to elucidate the mechanisms through which the endocannabinoid system impacts sleep physiology and the pharmacological properties of diverse cannabinoid agents concerning sleep modulation. More specifically, while promising therapeutic indications have surfaced, warranting further exploration, a pressing necessity exists for rigorous investigations into the safety and efficacy of cannabinoid therapies for sleep disorders, also aimed at determining the optimal dosing across diverse patient populations. Moreover, so far, there is a notable absence of published studies examining the impact of cannabis-based medicines on sleep among individuals diagnosed with physician-confirmed chronic insomnia disorder. As consumer interest in cannabis products grows and legal prescriptions become more widespread globally, there is a pressing need to comprehensively grasp the effects of these medications on sleep patterns and subsequent daytime functionality before integrating them into standard clinical practice. Future studies should employ validated assessments and prioritize cannabinoids or cannabinomimetics with favorable safety profiles and negligible abuse potential. In this context, while CBD has been proven not to induce dependence, researchers must ensure the high purity of CBD used in their studies and be vigilant for possible contamination with THC, although to overcome this impasse, future genetic engineering research can be accelerated to create strains of the cannabis plant with higher CBD content compared to the current one [103].

Within this framework, PEA, a compound completely devoid of toxic effects or abuse liability, emerges as a compelling candidate. Evening consumption of PEA has been shown to expedite sleep onset and enhance morning alertness, suggesting its potential utility in populations grappling with sleep onset difficulties and morning awakenings. Future research on PEA and sleep should, therefore, concentrate on such cohorts [58]. Future research on PEA must aim to uncover the intricate biochemical mechanisms linking the activity of peroxisome proliferator-activated receptors (PPARs) to the intricate transmitter systems that govern sleep architecture and the regulation of the sleep–wake rhythm. This endeavor gains particular relevance within the realm of phytocannabinoid action, given compelling evidence indicating its interaction with these pivotal nuclear sites.

Moreover, it is also relevant to note that there is preliminary evidence suggesting that the therapeutic properties of CBD and N-acylethanolamides may also involve epigenetic mechanisms, including DNA methylation, histone modifications, and regulation of miRNA expression [104]. This opens new perspectives for understanding the complex molecular and cellular mechanisms responsible for the effects of these compounds on sleep; therefore, further research is needed to delineate their complex pharmacodynamic profiles. The evolving understanding of the endocannabinoid system’s role in orchestrating the sleep–wake cycle, as underscored in this expert opinion, emphasizes the pressing need for continued inquiry. Although still in its early stages, research at the intersection of cannabinoids, N-acylethanolamides, and sleep presents a rich avenue for exploration. Advancing this field requires well-designed clinical trials featuring robust sample sizes and rigorous comparators (including both placebo and active treatment, where feasible), coupled with validated assessments encompassing measures of next-day function such as cognition and driving performance.

Furthermore, significant strides in the diagnosis and management of sleep disorders are on the horizon. Anticipated developments include simplified access to polysomnography and streamlined methods for sleep scoring, complemented by the integration of cutting-edge technologies such as artificial intelligence and telemedicine, all of which hold great promise for advancing the field of sleep pathophysiology research.

In conclusion, since insomnia is a disabling condition, research is committed to finding compounds that are characterized by low toxicity alongside high efficacy. Cannabinoids and acethanolamydes have shown very promising properties. Therefore, incorporating such compounds into comprehensive treatment approaches alongside CBT-I and lifestyle adjustments offers a holistic strategy for tackling the multifaceted nature of sleep disorders [105].

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
