# Peer review of "Cannabinoids and Sleep: Exploring Biological Mechanisms and Therapeutic Potentials"

_ijms, 2024, doi:10.3390/ijms25073603_

Round 1

Reviewer 1 Report

Comments and Suggestions for Authors

In the present review Authors addressed the current literature on the role of Cannabinoids on sleep regulation and the potential use  in treating sleep disorders.

Major comments:

in its current form, the manuscript is very difficult to read and, in many parts, extremely confusing.

I suggest a subdivision into paragraphs with an introduction that highlights the aim of the review followed by:

a brief description of the endocannabinoid system.

role of the endocannabinoid system in the regulation of sleep (endocannabinoid and exogenous cannabinoid modulation);

 At the end of this description, it will be important to highlight the potential use of for the treatment of sleep disorders.

Author Response

Rev1

Open Review

Quality of English Language

(x) I am not qualified to assess the quality of English in this paper
( ) English very difficult to understand/incomprehensible
( ) Extensive editing of English language required
( ) Moderate editing of English language required
( ) Minor editing of English language required
( ) English language fine. No issues detected

Yes

Can be improved

Must be improved

Not applicable

Does the introduction provide sufficient background and include all relevant references?

( )

( )

(x)

( )

Are all the cited references relevant to the research?

( )

( )

(x)

( )

Is the research design appropriate?

( )

( )

(x)

( )

Are the methods adequately described?

( )

( )

(x)

( )

Are the results clearly presented?

( )

( )

(x)

( )

Are the conclusions supported by the results?

( )

( )

(x)

( )

Comments and Suggestions for Authors

In the present review Authors addressed the current literature on the role of Cannabinoids on sleep regulation and the potential use  in treating sleep disorders.

We are grateful for the time you spent reviewing our article and for the valuable advice you provided to improve it

Major comments:

in its current form, the manuscript is very difficult to read and, in many parts, extremely confusing.

I suggest a subdivision into paragraphs with an introduction that highlights the aim of the review followed by:

Thank you for your suggestion. We have substantially modified the manuscript, not only by dividing it into paragraphs but also by restructuring it differently:

Endocannabinoid system

Endocannabinoids and sleep

Phytocannabinoids and synthetic cannabinomimetics

N-acylethanolamides

Research into cannabinods and sleep

Conclusion

a brief description of the endocannabinoid system.

Response: Thank you for your suggestion. We dedicated the first paragraph on the argument. In addition to dedicating and implementing the first paragraph, we have expanded the sections concerning the endocannabinoid system in the various subsequent paragraphs.

role of the endocannabinoid system in the regulation of sleep (endocannabinoid and exogenous cannabinoid modulation);

Response: Thank you for your suggestion we highlighted it the paragraph “Endocannabinoids and sleep” describing both mechanism of action.

 At the end of this description, it will be important to highlight the potential use of for the treatment of sleep disorders.

Response: Thank you for your suggestion. We have done so by emphasizing the importance of both endogenous and synthetic cannabinoids as a significant pharmacological perspective, both in monotherapy as demonstrated by the studies cited in the previous paragraphs, and as add-on therapy, particularly palmitoylethanolamide.

Submission Date

08 February 2024

Date of this review

01 Mar 2024 11:38:52

Reviewer 2 Report

Comments and Suggestions for Authors

1. The article should be divided into paragraphs/section. Otherwise, some of information is repeated in the text, and consequently, it is hard to follow. 

Obviously, the paper is submitted as an opinion, however, the Authors should try to to preserve the continuity of thought that is put down on paper

2. The manuscript should include both preclinical and clinical studies (detailed characterization and examples). Also, the Authors should focus on natural as well as synthetic compounds and should analyse them in depth in terms of potential mechanism of action, but also, if possible, demonstrate possible differences in the action of ligands

4. Please expand the abbreviation before its first use (e.g., AEA, 2-AG, OEA, etc.; line 32, 34, etc.)

Comments on the Quality of English Language

minor changes are required

Author Response

Rev2

Quality of English Language

( ) I am not qualified to assess the quality of English in this paper
( ) English very difficult to understand/incomprehensible
( ) Extensive editing of English language required
( ) Moderate editing of English language required
(x) Minor editing of English language required
( ) English language fine. No issues detected

Yes

Can be improved

Must be improved

Not applicable

Does the introduction provide sufficient background and include all relevant references?

( )

( )

( )

(x)

Are all the cited references relevant to the research?

(x)

( )

( )

( )

Is the research design appropriate?

( )

( )

(x)

( )

Are the methods adequately described?

( )

( )

( )

(x)

Are the results clearly presented?

( )

( )

( )

(x)

Are the conclusions supported by the results?

( )

( )

(x)

( )

We appreciate the time you dedicated to reviewing our article and the invaluable advice you offered to enhance its quality.

Comments and Suggestions for Authors

  1. The article should be divided into paragraphs/section. Otherwise, some of information is repeated in the text, and consequently, it is hard to follow.

Thank you for your suggestion. We have substantially modified the manuscript. We have divided the opinion into paragraphs, and by structuring it accordingly, we have eliminated repetitions and typos. The division was intended to make the article's reading flow smoother:

Endocannabinoid system

Endocannabinoids and sleep

Phytocannabinoids and synthetic cannabinomimetics

N-acylethanolamides

Research into cannabinods and sleep

Conclusion

Obviously, the paper is submitted as an opinion, however, the Authors should try to to preserve the continuity of thought that is put down on paper

  1. The manuscript should include both preclinical and clinical studies (detailed characterization and examples). Also, the Authors should focus on natural as well as synthetic compounds and should analyse them in depth in terms of potential mechanism of action, but also, if possible, demonstrate possible differences in the action of ligands

Thank you for your suggestions; we have incorporated them. We have devoted a section to cannabinomimetics, analyzing their mechanism of action and citing preclinical and clinical studies in the paragraph titled "Research into Cannabinoids and Sleep."

  1. Please expand the abbreviation before its first use (e.g., AEA, 2-AG, OEA, etc.; line 32, 34, etc.)

Thank you for your suggestion, we have corrected it.

Comments on the Quality of English Language

minor changes are required

Thank You, we did it

Submission Date

08 February 2024

Date of this review

28 Feb 2024 10:59:14

Reviewer 3 Report

Comments and Suggestions for Authors

The opinion received for review entitled - Cannabinoids and Sleep: Exploring Biological Mechanisms and Therapeutic Potentials about a very popular phytocannabinoid in relation to sleep, in my opinion, although it contains a lot of interesting information, is written very incoherently. There are no specific sections marked, such as: Introduction.... summary. It lacks systematicity in issues that should also be bulleted or titled to make the work easy to understand. Unfortunately, it also lacks figures that would logically show the impact of phytocannabinoids - advantages and disadvantages on the course of sleep. The table on pages 11 and 12 is very illegible, it does not contain specific data, only a copy of the text in a shortened version.

Author Response

Rev3

Open Review

Quality of English Language

(x) I am not qualified to assess the quality of English in this paper
( ) English very difficult to understand/incomprehensible
( ) Extensive editing of English language required
( ) Moderate editing of English language required
( ) Minor editing of English language required
( ) English language fine. No issues detected

Yes

Can be improved

Must be improved

Not applicable

Does the introduction provide sufficient background and include all relevant references?

( )

( )

(x)

( )

Are all the cited references relevant to the research?

( )

( )

(x)

( )

Is the research design appropriate?

( )

( )

(x)

( )

Are the methods adequately described?

( )

( )

(x)

( )

Are the results clearly presented?

( )

( )

(x)

( )

Are the conclusions supported by the results?

( )

( )

(x)

( )

Comments and Suggestions for Authors

The opinion received for review entitled - Cannabinoids and Sleep: Exploring Biological Mechanisms and Therapeutic Potentials about a very popular phytocannabinoid in relation to sleep, in my opinion, although it contains a lot of interesting information, is written very incoherently. There are no specific sections marked, such as: Introduction.... summary. It lacks systematicity in issues that should also be bulleted or titled to make the work easy to understand. Unfortunately, it also lacks figures that would logically show the impact of phytocannabinoids - advantages and disadvantages on the course of sleep. The table on pages 11 and 12 is very illegible, it does not contain specific data, only a copy of the text in a shortened version.

Response: Thank you for taking the time to review our article. As you will see, the manuscript has been substantially changed, and divided into sub-paragraphs to make the reading more coherent and fluid.

  1. Endocannabinoid system
  2. Endocannabinoids and sleep
  3. Phytocannabinoids and synthetic cannabinomimetics
  4. N-acylethanolamides
  5. Research into cannabinoids and sleep
  6. Conclusion

Furthermore, as you will read, we have decided, precisely due to the issue you raised, to insert a paragraph on "Phytocannabinoids and synthetic cannabinomimetics." The figure has been sent, but I believe it will be published as a graphical abstract. Regarding the table, since it is an opinion and not a review, we only wanted to summarize the studies and results cited in the study, and we have modified it.

Submission Date

08 February 2024

Date of this review

06 Mar 2024 09:20:02

Round 2

Reviewer 1 Report

Comments and Suggestions for Authors

The authors improved the quality of the manuscript as requested

Author Response

Thank you so much for reviewing our manuscript and for the advice that helped improve it.

Reviewer 2 Report

Comments and Suggestions for Authors

The paper has now been improved. However, I suggest to provide the Authors' opinion about recent and possibly further studies on cannabinoids and sleep in depth. Since it is an opinion, the Authors should provide theirs attitude to the work already done by other researchers. 

Are there any limitation which may be crucial for further studies in the opinion of the Authors?

Author Response

Thank you very much for the suggestion to comment on the work done so far,  to formulate hypotheses for future research in the field of the effect of cannabinoids on sleep, and their possible use in subjects with related disorders. We have incorporated all the very useful suggestions so that the last paragraph has been extended and changed. 
Title "Comments and Future Perspectives"
We liked the suggestion that made our work more complete.

Reviewer 3 Report

Comments and Suggestions for Authors

The work after the changes has been read. Now it looks much better.

Author Response

Thank you for the time spent reviewing our article and the comments that allowed us to improve it.